# The Emerging Role of Pericyte-Derived Extracellular Vesicles in Vascular and Neurological Health

**DOI:** 10.3390/cells11193108

**Published:** 2022-10-02

**Authors:** Kushal Sharma, Yunpei Zhang, Keshav Raj Paudel, Allan Kachelmeier, Philip M. Hansbro, Xiaorui Shi

**Affiliations:** 1Oregon Hearing Research Center, Department of Otolaryngology/Head & Neck Surgery, Oregon Health & Science University, Portland, OR 97239, USA; 2Centre for Inflammation, Centenary Institute and University of Technology Sydney, Faculty of Science, School of Life Sciences, Sydney, NSW 2007, Australia

**Keywords:** pericyte, PC-derived extracellular vesicle, blood–brain barrier, angiogenesis, neuroprotection, peripheral neuropathy, Parkinson’s disease

## Abstract

Pericytes (PCs), as a central component of the neurovascular unit, contribute to the regenerative potential of the central nervous system (CNS) and peripheral nervous system (PNS) by virtue of their role in blood flow regulation, angiogenesis, maintenance of the BBB, neurogenesis, and neuroprotection. Emerging evidence indicates that PCs also have a role in mediating cell-to-cell communication through the secretion of extracellular vesicles (EVs). Extracellular vesicles are cell-derived, micro- to nano-sized vesicles that transport cell constituents such as proteins, nucleic acids, and lipids from a parent originating cell to a recipient cell. PC-derived EVs (PC-EVs) play a crucial homeostatic role in neurovascular disease, as they promote angiogenesis, maintain the integrity of the blood-tissue barrier, and provide neuroprotection. The cargo carried by PC-EVs includes growth factors such as endothelial growth factor (VEGF), connecting tissue growth factors (CTGFs), fibroblast growth factors, angiopoietin 1, and neurotrophic growth factors such as brain-derived neurotrophic growth factor (BDNF), neuron growth factor (NGF), and glial-derived neurotrophic factor (GDNF), as well as cytokines such as interleukin (IL)-6, IL-8, IL-10, and MCP-1. The PC-EVs also carry miRNA and circular RNA linked to neurovascular health and the progression of several vascular and neuronal diseases. Therapeutic strategies employing PC-EVs have potential in the treatment of vascular and neurodegenerative diseases. This review discusses current research on the characteristic features of EVs secreted by PCs and their role in neuronal and vascular health and disease.

## 1. Introduction

Pericytes (PCs) are mural cells of the circulatory system that enshroud endothelial cells of capillaries, arterioles, and venules in the central and peripheral vasculature [1,2]. The pericytes play a vital integrative role in the CNS and PNS. In the brain, pericytes form an interface between the capillaries and brain parenchymal tissue and are integral to blood–brain barrier (BBB) integrity [3] and cell–cell communication between neurovascular units (NVUs) such as microglia, astrocytes, neurons, and endothelial cells [4]. As a component of the NVU, pericytes play a central role in regulating blood flow, vascular development, BBB integrity, neuroprotection, and neuroinflammation [4]. It is well established that PCs play a regenerative role in the CNS and PNS in pathophysiological processes such as ischemia [5,6,7], Alzheimer’s disease [4,8], and tumors [9]. However, the mechanisms through which the PCs exercise their multifaceted role are still unclear. In this review, we summarize the current advances in PC-derived EV research, mainly focusing on the properties of PC-derived extracellular vesicles and their role in maintaining neurovascular function in healthy and diseased states. As exosomes have emerged as an exciting new delivery vehicle on the therapeutic horizon, we also discuss how PC-derived exosomes may be employed in treating vascular and neurodegenerative disease.

### 1.1. PCs

Eberth first discovered PCs in the late 19th century; however, it was Rouget who made the first comprehensive description of the cell type [1,10]. Rouget noted the cells are embedded in the basement membrane of precapillary arterioles and post-capillary venules. In the 1920s, Zimmerman coined the word “pericyte” and classified different types of PCs based on morphology and position in the vascular network, also describing transitional forms such as vascular smooth muscle cells (VSMCs) [5,11]. Later studies identified several markers for PCs, including contractile proteins (myosin, smooth muscle actin, vimentin, desmin, and nestin), cell surface antigens (neuron-glial antigen 2 (NG2), a transmembrane chondroitin sulfate proteoglycan, platelet-derived growth factor receptor-β (PDGFR-β), alanyl aminopeptidase (CD13), a regulator of G-protein signaling-5 (RGS5), and a cell surface glycoprotein (MUC18 or CD146) [2,12]. Current studies of PCs often utilize genetic mice expressing a fluorescent protein under the control of NG2 (*Cspg4*) and *Pdgfrb,* such as NG2-dsRed, NG2/PDGFR-β-tdtomato, which labels several subpopulations of PCs [13]. Taking advantage of the Cre-Lox systems, pericyte function can be studied by knocking in or out the gene within the pericyte [14]. In addition, with advanced imaging techniques such as two-photon microscopy, researchers have investigated the fate of PCs under both normal and pathological conditions. Using PC depletion models, such as PDGFR-β+/− [15,16] and PC-specific Cre mice with a re-inducible diptherin toxin receptor (iDTR)—mice carrying Cre-dependent human diphtheria toxin receptor (DTR) [17,18] the consequences of PC ablation on neurovascular function have been determined. This has given us an in-depth understanding of the role of PCs in human health. In addition, the combined technique of optogenetic with 2-photon imaging has been shown to stimulate pericytes to generate a state of vessel constriction, eliciting pericyte-driven capillary diameter changes. However, this has been inconsistent in all studies [19], possibly due to the non-specificity of the Cre-reporter line or model/and or light source used. For instance, genetic mouse models such as NG2- and PDGFR-β reporter mice not only target the pericytes but also VSMC [13], fibroblast [20] or oligodendrocyte precursors cells. A specific genetic mouse model must be designed to target and validate the pericyte and its function. 

The origin of the PCs is heterogeneous and tissue-dependent [21]. The progenitor of PCs had long been thought to be mesenchymal stem cells (MSCs) [12]. For example, the origin of PCs in the liver [22], gut [22,23], and lungs [24] has been traced to the mesothelium, a single layer of the squamous epithelium [12]. However, in later genetic lineage tracing experiments, some of the face, brain, and thymus PCs were shown to have originated in the neural crest [25]. More recently, pericyte-like cells could be derived from human pluripotent stem cells [26]. Increasing evidence shows the PCs, like MSCs, transdifferentiate into various cell subtypes, including chondrocytes [27], adipocytes [27], osteoblast [28], phagocytes, neural progenitor [29], vascular cells [29], and microglia [30] both in vitro and in vivo.

### 1.2. Extracellular Vesicles

Extracellular vesicles (EVs) were first identified in cancer research. However, the role of EVs is now considered broader, that of a critical player in cell-to-cell communication [31,32]. EVs are micro- to nano-sized membrane-bound cell vesicles released into the surrounding field upon fusion of multivesicular bodies and the plasma membrane, which are then delivered into recipient cells. Previously, they were classified based on their biogenetic and biophysical properties into three main categories: microvesicles (100 nm to 1 μm), exosomes (50 to 150 nm), and apoptotic bodies (1–5 μm) [33]. However, more recent studies show a broader size range of EVs, not limited to previously reported classes [33,34]. Recent EV classes include exomeres (<50 nm), small exosomes (60–80 nm), large exosomes (90–120 nm), microvesicles (0.1–1 µm), migrasomes (0.5–3 µm), and oncosomes (1–10 µm). In general, exosomes are identified by multiple unique markers: tetraspanins, CD9, CD63, CD81, endosomal protein tumor susceptibility gene 101 (TSG101), and ALG-2-interacting protein X (ALIX) [33]. However, recent International Society of Extracellular Vesicles guidelines does not propose molecular markers for the characterization of EV subtypes [35,36,37]. A more thorough investigation of EV characteristics and physiology is needed to parse apart different subpopulations. In this review, we use the term “EV” to refer to all extracellular vesicles released by PCs, including exosomes and micro-vesicles. 

EVs arise from late endosomes formed by the inward budding of the multivesicular body (MVB) membrane [38,39]. Details of EV formation are shown in Figure 1. Briefly, invagination of the late endosomal membrane results in intraluminal vesicles (ILVs) of large MVBs [38,39]. The late-stage MVBs fuse with the cell membrane to release EVs by budding the plasma membrane. The released exosomes exhibit cell membrane proteins such as CD9, CD63, and CD81 on their surface, and these serve as markers for identification. EVs release involves two mechanisms: in one, an endosomal sorting complex is required for transport (ESCRT), and the other is ESCRT independent. ESCRT-dependent formation of exosomes involves multiple steps, including recognition and sequestration of ubiquitinated proteins to specific domains of the endosomal membrane via ubiquitin-binding subunits of ESCRT-0, formation of an ESCRT (I, II, III) protein complex to promote the budding process, and dissociation of the ESCRT I-III complex to form ILVs [38,40]. The ESCRT independent mechanism requires sphingomyelinase II (N-SMase) to catalyze the production of sphingolipid ceramide. The sphingolipid ceramide is necessary for forming exosomes, as it promotes the budding of intraluminal vesicles into MVBs [38,40,41,42]. Released EVs stored in the extracellular space have been shown to carry cargoes of bioactive compounds [38,42,43]. Once released in the extracellular space, EVs are distributed in different tissues, organs, and biological fluids, both locally and systemically [42]. Systemic exosomes are rapidly cleared from the blood circulation by macrophages, neutrophils, and endothelial cells and are transported to the liver, spleen, lungs, and gastrointestinal tract [44,45]. EV access to tissue involves multiple cellular uptake and release cycles [46]. However, different factors such as cellular origin, membrane composition, EV size, and pathological conditions in the host may affect how EVs are transported [47]. Although all cell types share in non-specific uptake of exosomes [48], specific targeting of recipient cells is required to deliver specific cargo and exert specific function [49]. The surface composition of the exosome mediates this delivery specificity. EV–target cell interactions involve tetraspanins, integrins, ECM proteins, immunoglobulin superfamily members, proteoglycans, heat shock proteins, and lectins [50]. For example, heat shock protein 70 predominantly clusters around the exosomal membrane [51]. There is evidence that it interacts with Toll-like receptor 4 to initiate a critical signaling cascade necessary for neuron survival [52]. Umbilical cord blood-derived exosomes express tumor antigens such as MHC-I, MHC-II, and tetraspanins (CD34, CD80) and stimulate T cell proliferation to produce antitumor activity [53]. EVs then interface with neighboring cells through different modes, including receptor-binding without internalization, phagocytosis, macropinocytosis, internalization by clathrin-caveolae- and lipid raft-mediated endocytosis, filopodia-based endocytosis, and bursting in the acidic environment [54]. If internalized, the EVs release their cargo in the cytoplasm, and the internalized cargo may regulate the cell at the transcription or translation level [32,55]. EVs carry a range of cargo for deposition into recipient cells, from genetic materials (RNA, DNA, and miRNA), to proteins, lipids, and transcription factor receptorss. They can effectively alter biological response. The exosome-mediated response could be either therapeutic or disease-promoting [56]. While it is clear exosomes are a promising therapeutic tool for treatment of different pathologies, it is clear they are also associated with development of diseases such as diabetes, Alzhemier’s, progression of cancer, and inflammatory disease [57].

### 1.3. PC-Derived EVs (PC-EVs)

#### 1.3.1. Identification of PC-EVs

PC-released EVs were first reported by Gaceb et al. in 2018 [58]. However, EV size and morphology were not well characterized until more recently with two studies on brain PCs [58,59,60]. PC-derived EVs are now well described by their size and distinctive markers [59,60]. The diameter of brain PC-derived EVs varies between 30 and 350 nm (standard EVs are 100 nm to 1 μm). Under transmission electron microscopy (TEM), brain PC EVs are shown double-layered, either spherical or cup-shaped, expressing EV markers such as CD9 and CD81. Consistent with the morphology identified in brain PC-EVs, PC-EVs from retinal blood vessels are also spherical-shaped, but the retinal EVs range in size between 100 nm and 1 μm. Most recently, Yin et al. (2022) extracted nanoparticle-like EVs from cavernous PCs, and these also had a cup-shaped morphology [61].

In general, PC-EVs are identified based on their expression of PC markers such as NG2, PDGFR-B, and a-SMA. In addition to their expression of the EV markers mentioned above and size, some of the PC-EVs also present with other markers such as tsg101, flotillin-2, actinin-4, Alix, TSG1010, CD9, CD63, and CD81 [60,61,62]. For example, EVs released from muscle-derived PCs express markers for CD9, CD63, and CD81. The size of purified EVs can be as small as 27 nm–50 nm, such as is the case of PC-EVs derived from cavernous PCs [61]. 

#### 1.3.2. PC-EV Cargo

Growth Factors
Gaceb et al., 2018 were the first to report brain PCs to secrete a variety of growth factors under different conditions [58]. Under normal unstimulated conditions, the brain PC-EVs carry insulin-like growth factor-binding protein (IGFBP), neurotrophin 3 (NT3), heparin-binding EGF-like growth factor (HB-EGF), brain-derived neurotrophic factor (BDNF), fibroblast growth factor (FGF), neuron growth factor (NGF), and vascular endothelial growth factor (VEGF) [58]. They found the growth factors released by PCs primarily through EVs, not through direct secretion. For example, PCs were found to directly release 1.5 pg/mL, 25 pg/mL, and 1 pg/mL, respectively, of bFGF, BDNF, and NGF, and the corresponding share of the EV release was1.5 pg/mL, 20 pg/mL, and 0.6 pg/mL of the growth factors [58]. This is suggestive that the PCs primarily release growth factors through EVs. Grace et al. found a 3–4 fold increase in EV growth factors when PCs are stimulated with PDGF-BB [58,63]. In addition to carrying vascular and neuronal growth factors, the PC-EVs also carry pleiotrophin (PTN) [18] and connective tissue growth factor (CTGF) [64].
Inflammatory Cytokines

Under pathological conditions, PCs can secrete proinflammatory factors such as interleukins (ILs), tumor necrosis factor-alpha (TNF-a), and interferon gamma-induced protein 10 (IP-10) [58]. Stimulating PCs with LPS leads to a higher release of cytokines and chemokines [58,65,66,67,68]. For example, LPS-stimulated PCs release EVs with inflammatory cytokines such as IL6, IL8, MCP, and IL-10 [58,63,69]. By contrast, stimulation with PDGF-BB leads to a lower concentration of cytokines in the PC-EVs than LPS stimulation [58]. The PCs also release anti-inflammatory mediators such as leukemia inhibitory factor (LIF) and heme oxygenase-1 (HMOX-1) in the disease state ([58,63,69]). Studies show the type of compounds released in EVs by PCs is highly dependent on the environment and functional state of the tissue [58,63,69]. The EVs may be contributing to condition-dependent inflammatory or anti-inflammatory action.
miRNA and Circular RNA

PC-EVs also carry microRNA (miRNA) [60]. miRNAs are 18–25 nucleotide lengths of non-coding RNA. The miRNAs bind to 3′ or 5′ untranslated regions of genes and regulate their post-transcriptional expression [60,70,71]. The miRNAs also have autocrine effects when released into the extracellular space and transported to other body parts by blood or urine [72]. The miRNAs play an important role in various cellular events, and their dysregulation is linked to the progression of the disease [73]. The EV-derived miRNAs have been reported to play a therapeutic role in a wide array of pathology, including cancer [74], diabetes [75], neurodegenerative disease [76], hypertension [77], cerebrovascular disease [78], osteoporosis [79], and atherosclerosis [80]. Wu et al. demonstrated that brain PC-EVs carry significantly different miRNA in normal animals than in spontaneously hypertensive rats. In the study, 386 different miRNAs were identified in the normotensive rats, while 225 different miRNAs were identified in the hypertensive animals. miRNAs such as miR-26a, miR-143, miR 122-5p, miR-6240, miR-122-5p miR-11980, miR-181a, miR-21, miR-29a, let-7c-5p, let-7a-5p, and miR-1285 were differentially expressed in the two conditions [60]. Although the function of the EV-miRNA is well identified in specific systems, further study is needed on the role of miRNAs carried by PC-EVs in vasculogenesis, neuro-regeneration, and tissue and organ regeneration.

Circular RNA (circRNA) is a stable and ubiquitous form of head-to-tail spliced coding or non-coding RNA. There is substantial evidence that circRNAs are modulators of physiological processes and cell signaling pathways in the progression of vascular disease, cancers, and neurological disorders [81,82]. Liu et al. reported that PC-derived EVs from db/db mice shed the circRNA, cPWWP2A, affecting PC coverage of endothelial cells and vascular integrity [83]. Ye et al. identified hypoxia upregulated circEhmt1 expression in PCs and demonstrated that circEhmt1 was transferred from PCs to endotheliocytes via EVs [84]. Overexpression of circEhmt1 prevents endotheliocyte damage from exposure to high glucose in an in vitro model [84]. Other investigations have shown a crucial role for the circEhmt1-mediated nuclear factor I-A/NOD-like receptor family pyrin domain-containing protein 3 NFIA/NLRP3 signaling in retinal microvascular dysfunction. This signaling pathway could be a promising target for managing diabetic retinopathy [84].

## 2. PC-EV Physiology

PCs are at the interface between blood and parenchyma [85]. They interact with neighboring cells and work synergistically to regulate blood flow, maintain the blood–brain barrier, and regulate angiogenesis and neuronal growth by releasing various growth factors [86]. PCs are also first in line to sense local environment changes, such as hypoxia, inflammation, pathogens, and high blood glucose [87]. In response to these changes, PCs secrete various inflammatory mediators (cytokines, adhesion molecules, chemokines, interleukins, and extracellular vesicles) and extracellular matrix [58,63,69]. These secreted molecules communicate in a paracrine way with nearby cells and regulate pro- or anti-inflammatory function, angiogenesis, neuroprotection, BBB integrity, and tissue regeneration [63]. The stimuli also enhance production of EVs which contain signaling molecules involved in intermediate and long distance signaling. PCs in the human brain release EVs in both the unstimulated and stimulated condition (listed in Table 1). However, the level of EV release after PC stimulation (LPS or PDGF-BB) is remarkably higher than in the unstimulated condition [58]. Like EVs derived from stem cells, PC-EVs may also be a therapeutic candidate useful in management of neurodegenerative disease. For example, the PC -EVs have been demonstrated to carry circular RNA in response hypoxia [88] and diabetes-related stress [83]. This leads to discussing the PC-EV secretome and its role in vascular and neuronal health.

### 2.1. Angiogenesis

PC-EV regulation of angiogenesis was first documented in 2018. PCs and ECs interact in a paracrine manner with different extracellular factors to affect angiogenesis [89]. Studies have shown the EVs released by PCs contain a variety of angiogenic-promoting factors such as VEGF, CTGF, PLGF, FGF, microRNA, and circRNA [58,60,63,64,69,84]. Figure 2 illustrates some of the postulated mechanisms through which PC-EVs modulate angiogenesis. PC-EV-derived CTGF is mainly involved in the initial stages of angiogenesis by activating the ERK1/2-STAT3 axis in vascular endothelial cells (VEC) [64]. CTGF also modulates other angiogenetic factors such as VEGF, ANG-2 (angiopoietin-2), and matrix metalloproteinases (MMPs) [64,90,91]. However, whether other angiogenic factors are functionally active in PC-EVs is yet to be determined. However, the angiogenic role of these molecules is well demonstrated in non-PC-derived EVs, including platelet-derived EVs, MSC-derived EVs, and cancer-derived small EVs [89]. VEGF and FGF released by platelet-derived EVs act on the VEGF and FGF receptors to exert a pro-angiogenic effect by activating PI3 kinase, src kinase, and ERK1/2 [89,92]. Studies also show that VEGF and FGF act cooperatively to promote proliferation, migration, and EC tube formation through a PI3K pathway [92,93]. EVs derived from MSCs containing VEGF and FGF promote angiogenesis through a nuclear factor-κB pathway [89]. EVs from hypoxia-treated human adipose-derived stem cells carry VEGF and have been shown to enhance angiogenesis in the grafted tissue through VEGF/VEGF-R signaling [94]. Earlier studies reported that VEGF tumor cell release is associated with EVs [95,96]. Cancer-derived small EVs contain an isoform of VEGFA, which can induce endothelial cell migration and tube formation [97,98]. EV-derived VEGF-A is shown to increase angiogenesis and permeability in endothelial cells in the brain [89]. Recent studies of VEGFs indicate that VEGF isoforms are bound to the surface of small EVs (sEVs) through heparin surface binding or bound to microvesicles through heat shock protein 90 (HSP90) [97,99]. Microvesicle-associated VEGF-HSP90 has been shown to stimulate VEGFR2 phosphorylation in endothelial cells and tube formation [100]. EV-associated VEGF189 has also been shown to stimulate the cellular domain of VEGFR2 and induce phosphorylation. The VEGFR2 signaling elicited in endothelial cells then increases microvessel density in vivo. This finding supports the idea that VEGF in PC-EVs may interact with the extraceullar domain of VEGFR2 and the VEGFR signaling induces angiogenesis. Several miRNAs have been identified in the PC-EVs of brain microvascular PCs, including miR26a, miR-143, miR122-5p, miR181a, miR21, miR29a, Let-7 family, etc. [60]. Although the function of these miRNAs released from PC-EVs is yet to be determined, they have been shown to stimulate angiogenesis in other cell-derived EVs. For example, miR26a shed by glioblastoma stem cell-derived EVs promotes proliferation, migration tube formation, and angiogenesis by activating a PTEN/PI3/Akt pathway [74]. Similarly, miR-21 released by endothelial colony-forming cell-derived EVs (ECFC) stimulated EC proliferation, migration, and tube formation via PI3/AKT pathway [80]. 

The microenvironment has a large influence on the functionality of the PC secretome [58,84]. In particular, hypoxic conditions set the stage for PCs to stimulate a pro-angiogenic secretome [84,88]. In vitro studies show that hypoxia-stimulated PC-EV secretion promotes angiogenesis and wound healing by promoting growth factors, DNA, and circular RNA to modulate the angiogenic program in response to microenvironmental changes [84,88]. Ye et al., for example, identified hypoxia upregulated circEhmt1 expression in PC-EVs under high glucose conditions. The circEhmt1 expression protected the endotheliocytes against high-glucose damage and promoted angiogenesis through upregulation of NFIA, followed by inhibition of NLRP3 [84]. Similarly, high-glucose treated PCs released PC-EVs containing cPWWP2A, which increased the capacity for proliferation, migration, and tube formation in ECs [83]. The studies also revealed the critical role of PC-EV signaling in retinal microvascular dysfunction and suggested signaling influenced by circRNA could be therapeutically targeted in treating diabetic retinopathy [83,84].

### 2.2. Blood–Brain Barrier (BBB)

The BBB comprises a complex molecular structure that, as an ensemble, regulates the extracellular environment of the CNS [101]. Endothelial cells, astrocytes, PCs, and extracellular matrix are critical for the formation and maturation of the barrier [101,102]. PCs are a central element of the vascular and neuronal unit [4] and, as such, are critical players in maintaining BBB integrity and vessel stabilization [103]. PCs stabilize the BBB by releasing signaling factors that determine the expression of tight junction protein in endothelial cells and regulate the rate of bulk-flow transcytosis of fluid-filled vesicles across the BBB [104]. In addition, PC transport of substances across the BBB includes the clearance of toxins from the brain [85]. Loss of PCs results in loss of tight junctions between ECs, leading to BBB breakdown [85,105]. Studies demonstrate that neurodegenerative diseases such as Alzheimer’s disease, Parkinson’s disease, Huntington’s disease, dementia, and amyotrophic lateral sclerosis are associated with BBB breakdown. Targeting the effect of PCs on the BBB could control the course of some neurological disorders [106,107,108]. How do PC-EVs preserve BBB integrity in both the normal and pathological state?

A recent study has demonstrated that PC-EVs protect the microvascular endothelial cell barrier under hypoxic conditions [59]. The PC-derived EVs decreased the permeability induced by the hypoxia by increasing the expression of tight junction (TJ) protein, including endothelial zonula occludens-1 (ZO-1) and claudin 5 [59]. Proteomic analysis of PC-EVs has identified Ang-1 in the vesicles, suggesting PC-EVs are a major release vehicle [64]. Angiopoietin-1 (Ang-1), a key agent necessary for BBB integrity, is continuously secreted by PCs [109]. Ang-1 ligation stimulates the Tie2 receptor, followed by activation of downstream signaling via the phosphatidylinositol 3-kinase (PI3K)/Akt pathway, conferring resistance to permeability [110]. It has been proposed that Ang-1/Tie-2 mediates endothelial maturation and stability and reduces vascular leakage [111,112]. The Ang1/Tie-2 upregulates the expression of junction proteins such as occludin (Ocln) and VE-cadherin and thus stabilizes endothelial cells [109]. Although the molecular mechanism of action is well understood, the release mechanism is yet to be identified. Studies have suggested that Ang-1 is released through extracellular vesicles [113]. Ang-1 shed by PC-EVs could be a potential new therapeutic target for intervention in several pathologies associated with BBB disruption (Figure 3).

PC growth factors shed by PCs, including VEGF, play a crucial role in maintaining the integrity of the BBB and neurovascular unit (NVU) [114]. However, the exact role of the VEGF derived from PC-EVs is unknown. Recently, it was shown that glioblastoma multiforme (GBM) cells located in the NVU in close contact with the BBB secrete EVs enriched in vascular endothelial growth factor-A (VEGF-A) [115]. Interestingly, the GBM-secreted EVs (GBM-EVs) contained VEGF-A, promoted angiogenesis, and increased BBB permeability by reducing the expression of tight junction proteins such as claudin-5 and occludin in endothelial cells [116]. By contrast, VEGF-B shed by MSC-EVs acts as a pro-survival factor by decreasing vascular leakage and promoting microvessel growth in ischemic tissue [117]. Distinguishing the subtypes of VEGF in PC-EVs will be necessary for understanding the exact role of VEGF in regulating the BBB.

MiRNAs also modulate the integrity of the BBB by regulating PC coverage-associated molecules [118]. One example is the miR-27 shed by PC-EVs [60]. While miR-27a targets VE-cadherin to compromise BBB integrity, miR-27b promotes the interaction of endothelial cells with PCs by targeting semaphorin 6A/D (SEMA6A/D), leading to a strengthening of the endothelial barrier [119]. PC-EVs release several miRNAs, but released levels can be abnormally high under pathological conditions. Understanding the different roles of miRNAs derived from PC-EVs will be necessary if therapeutic applications are developed against pathologies associated with BBB disruption. PC secretion of MMPs, chemokines, and adhesion molecules (ICAM-1, E-selectin, and VCAM-1) is well characterized, and these molecules are believed to be regulating the BBB [58,63,69]. However, their release through PC-EVs still needs to be better characterized and studied. Further investigation is required to identify other BBB regulating molecules in PC -EVs and understand their role in the physiologically normal and pathological state. The list of diseases in which aberration of the BBB is recognized as a primary cause includes asthma, diabetic retinopathy, stroke, and neurodegenerative disease (Alzheimer’s disease, Parkinson’s disease, Huntington’s chorea, amyotrophic lateral sclerosis, and multiple sclerosis) [120]. It is foreseeable that PC-EVs might be used in a therapeutic role in maintaining BBB integrity.

### 2.3. Neuron Health

PCs protect neurons by protecting the endothelium, stabilizing the BBB, and releasing neurotrophins [121]. PC death in pathological states results in BBB damage, often followed by neuronal death and progression to neurodegenerative disease (ND) [120]. Promoting neuronal survival is one of the key strategies in preventing ND. Recovery of neuronal injury was recently shown to be associated with PC-EV mediated neuroprotection and neurogenesis. Although PC-EV constituents were not identified, the protective mechanism was shown associated with enhanced Bcl2 (anti-apoptotic) expression and inhibition of the Bax apoptotic pathway [59]. In some instances, PC-EVs promoted neuron survival through phosphorylation of Akt and eNOS and inhibited the JNK/jun-c cell death pathway [64]. The expression of neurotrophic factors like BDNF, NT-3, and NGF also promoted neurogenesis [64]. However, these results raise a question regarding the molecular pathways involved in PC-EV-induced neuroprotection and neurodegeneration. It is well known that PCs release several neurotrophic factors which accompany neuroprotection and neuro-regeneration [58,63]. BDNF is, among them, a critical neurotrophic factor responsible for the development of the nervous system. The BDNF is shown to have a role in neurogenesis, neuroprotection, neurodegeneration [122], synaptic plasticity [123], and resistance to neuronal stress [124]. The mechanism of BDNF release from PCs, including its associated pathway, is not fully understood. BDNF has been identified as one of the neurotrophic growth factors released in PC-EVs [58]. The EV release mechanism enables BDNF to reach recipient cells (neurons), targeting different intracellular compartments such as mitochondria, cytoplasm, and endoplasmic reticulum to produce its neuroprotective effect. Indeed, BDNF-specific binding to its receptors would have a strong neuroprotective effect. We hypothesize that BDNF released from PC-EVs interacts with tropomyosin receptor kinase B receptors. The BDNF is internalized upon ligand binding and then stimulates neurite outgrowth, plastic behavior, and survival through MAPK, PLCy, and PI3K pathways [125] (Figure 4). These pathways target the transcription factor CREB, which enhances BDNF gene expression and further promotes survival, differentiation, and neurogenesis [125]. Clinical and preclinical experience suggests this ubiquitous growth factor plays an essential role in schizophrenia [126], addiction [127], Rett syndrome [128], as well as other psychiatric and neurodevelopmental diseases [129,130]. The function of neurotrophic factors in PC-EVs, including BDNF, is yet to be determined. However, the neuroprotective and neurodegenerative roles of BDNF are well demonstrated in other cell-derived EVs, such as MSC-derived EVs. MSC-derived EVs attenuate severe intraventricular hemorrhage (IVH), and this protection is essentially mediated by BDNF transferred via EVs [131]. Exosome-delivered BDNF might be used as a therapy to attenuate hypoxia/reoxygenation (H/R)-induced apoptosis [132]. The BDNF inhibits oxidative stress and helps maintain mitochondrial membrane potential in brain cells damaged by ischemia or reperfusion (I/R) [132]. Neurodegenerative diseases, including Parkinson’s disease and major depressive disorder (MDD), are linked to alteration in exosomal BDNF [133,134]. Treatment of MDD with antidepressant drugs increases exosomal BDNF and reduces MDD [134]. Since an adequate level of BDNF is necessary to prevent neurodegenerative disease, PC-EVs rich in BDNF might be used as a therapeutic strategy. PC-EVs also release other growth factors such as bNGF, GDNF, PLGF, and PTN which also might be utilized in treatment of neurodegenerative disease [58,64]. For instance, GDNF has been shown to be the most potent neuroprotective and neurodegenerative agent in PD [135]. A study has demonstrated that EVs loaded with GDNF have a strong neuroprotective effect [136]. Similarly, EVs loaded with the neuroprotective agent NGF delivered the agent into an ischemic region and ameliorated neuro-inflammation, reduced cell death, and promoted neurogenesis [137]. Growth factors like PTN [18,138] and PLGF [138] show promising neuroprotective effects against neuron loss. Delivered in EVs they might be useful in providing protection against neurodegenerative disease.

PCs respond to inflammatory conditions by releasing a subset of inflammatory mediators that facilitate cell survival, regeneration, or inflammation [63,69,139]. Secretome analysis shows PC- EVs to release different cytokines, including Il-6, Il-10, Il-8, or MCP-1, in both the unstimulated and stimulated conditions [58,63]. Studies demonstrate that these inflammatory mediators may play a role in neuroprotection. For instance, IL-6 facilitates motor neuron survival in preclinical animal models of motor neuron disease by enhancing BDNF release [140,141]. Damaged peripheral nerves are repaired [142,143], and IL-6 affects the long-term potentiation activity of neurons [144]. Similarly, PC secretion of IL-10 is known for its immunoregulatory and anti-inflammatory activities. The BBB is strengthened by inhibiting the impairment of tight junctions and downregulating claudin-4 expression [145]. The secretion of IL-10 could also have favorable effects in other states, such as neurological disorders and inflammation. When blood vessel integrity is affected, PC-EV secretion of inflammatory cytokines such as MCP-1 and IL-8 may also harm health [146,147]. MCP-1 is implicated in several CNS inflammatory states, such as stroke [148], meningitis [149], and multiple sclerosis [150,151]. It regulates brain endothelial cell permeability in vitro by altering tight junction (TJ) protein [150]. IL-8 is also involved in CNS diseases such as meningitis [152] and atherosclerosis [153]. The Il-8 increases BBB permeability and, in the latter, participates in the production of atherosclerotic plaque. Lastly, miRNA could be the target for PC-EV-mediated neuroprotection. miR29a, miR15b, and miR200, identified in PC-EVs [60], are believed to contribute to neuroprotection through inhibition of apoptotic signaling [154]. While PC-EVs release many miRNAs, the function of these miRNAs is, in general, unknown. Investigating the role of miRNAs derived from PC-EVs is needed if their therapeutic potential in treating neurodegeneration is to be unlocked.

## 3. Therapeutic Potential of PC-EVs

Ongoing EV-based clinical trials investigate EVs as diagnostic, prognostic biomarkers, and therapeutic agents for various diseases, including neurodegenerative and vascular diseases. Mesenchymal stem cells (MSCs), dendritic cells (DCs), and tumor cell-derived EVs are under clinical trial for their therapeutic applications. While most clinical trials use these sources for therapeutic EVs, a rapidly increasing number of studies are also investigating pericytes as primary sources of therapeutic EVs. Here, we proceed to describe the potential role of PC-EVs in treating neuronal and vascular disease.

### 3.1. CNS Disease

#### Parkinson’s Disease

Parkinson’s disease is a brain disorder associated with a dopaminergic deficiency that causes unintended and uncontrollable movement [155]. Although no cure for Parkinson’s disease is known, medication, surgical treatment, and other therapies can often relieve some of its symptoms. Growth factors such as GDNF have been shown to promote neuroprotection from the toxic insult and regeneration of neurons damaged by Parkinson’s disease [156,157]. GDNF treatment protected and restored DA neurons in rodent and aged primate models of PD [156,158,159]. Earlier studies used genetically modified autologous macrophages for active targeted delivery of GDNF through EVs to preserve DA neural degradation [159]. These results demonstrate the promise of regulating BDNF, GDNF, or other growth factors secreted in PC-EVs. These results suggest novel therapeutic strategies to promote brain or peripheral nerve regeneration and treat Parkinson’s disease. The administration of PDGF-BB has recently been investigated in phase 1/2a clinical trials against Parkinson’s disease [135]. PCs under stimulation of PDGF-BB secretome induce neuro-restorative mechanisms in Parkinson’s disease and partially restore tyrosine hydroxylase positive nerve fibers by improving dopamine-transporter binding. The effect is mediated by normalizing abnormal vasculature in Parkinson’s disease [136]. PDGF-BB activation may release multiple growth factors via PC-EVs that support vascular stability and neuroprotection. Growth factors, including BDNF and GDNF, have shown promise in enhancing the survival of dopaminergic neurons and improving dopaminergic neurotransmission and motor performance [137].

### 3.2. Peripheral Neuropathy

#### 3.2.1. Spinal Cord Injury

Spinal cord injury (SCI) often leads to paralysis, and therapies to treat SCI are still challenging for researchers and physicians (references). Transplantation of mesenchymal stem cells (MSCs) is a promising treatment for SCI, but problems like immunological rejection and tumor formation are still to be resolved. Some studies have shown that EVs derived from MSCs protect the SCI by enhancing the survival of PCs and improving the integrity of the blood–spinal cord barrier. Yuan et al. first described the protective role of EVs derived from PCs in SCI [59]. In the study, PC-EVs were transplanted into mice with SCI to study the restoration of motor function and explore the underlying mechanism.

Interestingly, PC-EVs reduced pathological changes, improved motor function, and increased blood flow and oxygen to the damaged area after SCI. In addition, PC-EVs enhanced endothelial function, protected the blood-spinal cord barrier, and reduced edema [59]. The mechanism entailed attenuating the expression of HIF-1a (endothelial survival), Bax (apoptosis), Aquaporin 4 (edema), and MMP2 (BBB) [160]. The treatment also increased the expression of claudin 5 (tight junctions) and bcl-2 and inhibited apoptosis. A corresponding in vitro study showed that PC-EVs could protect the endothelial blood-spinal cord barrier and endothelial cells under hypoxic conditions through a PTEN/AKT pathway.

#### 3.2.2. Erectile Dysfunction

Studies have shown that PCs may promote axonal regeneration and barrier function in a mouse model of erectile dysfunction [161]. Yin et al. demonstrated that PC-derived extracellular vesicle-mimetic nanovesicles (NV) extracted from cavernous mouse tissue significantly improved erectile dysfunction and recovery of the peripheral nerve injury [61]. PC-NVs significantly promoted angiogenesis and Schwann cell migration, pelvic ganglion neurite sprouting, and improved dorsal root ganglion movement. Treatment with PC-NVs significantly ameliorated erectile dysfunction and provided functional recovery of the peripheral nerve injury, improving motor and sensory function [62]. The mechanism is associated with cell survival signaling (Akt and eNOS) and enhanced expression of neurotrophic factors (BDNF, NT-3, and nerve growth factor).

### 3.3. Diabetic Retinopathy

Diabetic retinopathy (DR) is a complication of diabetes that can cause vision loss and blindness. Pathological DR-associated changes include PC loss, impairment of the blood-retina barrier, endothelial dysfunction, and basement membrane thickening [162,163]. Liu et al. recently reported that diabetes-related stress upregulates circRNA expression, such as cPWWP2A, in PC-derived EVs [83]. An in vivo study has shown the promise of cPWWP2A overexpression in alleviating retinal vascular dysfunction [82,83]. The cPWWP2A alleviated diabetes mellitus-induced retinal vascular dysfunction by inhibiting miR579 activity. The miR579 inhibition leads to higher expression of Ang-1, occludin, and SIRT1. The study also revealed a novel mechanism by which exosomal circRNAs such as circEhmt1 play a protective role in the DR that accompanies retinal microvascular dysfunction. The studies are suggestive PC-EVs carrying circRNA could be targeted to activate signaling pathways to treat diabetic retinopathy.

### 3.4. Infection

Sepsis is an ever-present risk of severe disease which can accompany infection. Treatment strategies are limited. PC-derived EVs have been reported to be therapeutically useful in cases of sepsis [64]. PC-EVs can protect vascular endothelial cells (VECs) from injury through exosomal delivery of CTGF from PCs to VECs. The CTGF protects pulmonary vascular function in sepsis by promoting proliferation and angiogenesis through activation of the CTGF-ERK1/2-STAT3 pathway. These studies suggest PC-derived EVs might be used to target and activate signaling pathways as an adjunct in treating infection [64].

### 3.5. Muscle Atrophy

Limb immobilization for an extended period in the elderly can result in muscle atrophy [164,165]. Earlier studies in mice had suggested that prolonged oxidative stress during immobilization reduces the recovery capacity of skeletal muscles. In such cases, the muscle-resident PCs become dysfunctional and lose antioxidant defense with disuse. Wu et al. demonstrate that H_2_O_2_-primed PC-EVs improve the recovery of young and aged muscles after prolonged inactivity (reference here). Proteomic analysis of the EVs differentially detected regulators of proteins associated with ECM and anti-inflammatory and antioxidant processes [166]. This study demonstrated that PC-EVs could also stimulate signaling pathways to reduce oxidative stress and restore ECM, indicating utility in therapeutic strategies against disease and aging-related oxidative stress.

## 4. Conclusions and Future Perspectives

PC-EVs are a recent discovery. Mechanisms underlying PC-EV-related vascular-neuronal dysfunction are not yet known; the therapeutic potential of PC-EVs has not been validated for many neurodegenerative disorders. Vascular injury-induced inflammatory cytokine release could damage vascular and neuronal tissues. However, PC-EVs promise new therapeutic approaches and many new options for managing the neurodegenerative disease associated with vascular damage. Some of these are shown in Figure 5. PC-EVs released under physiological conditions mediate their effect through the cargo they carry. The cargo includes angiogenic growth factors (FGF-2, VEGF) and neurotrophic growth factors (BDNF, NGF, and GDNF) [58]. The proteins carried by PC-EVs have a range of effects. These include activating signaling pathways, promoting the survival of PCs, ECs, and neurons, regulating neurogenesis and angiogenesis, and contributing to synaptic plasticity and cognitive function. Loss of PCs in various disease states can disrupt the blood–tissue barrier, cause endothelial cell dysfunction, or lead to neurodegeneration [167]. Treatment with PC-EVs can be utilized in therapeutic approaches to prevent vascular or neurological damage associated with neurodegenerative diseases such as Alzheimer’s disease (AD) by their effects on cell survival, angiogenesis, BBB integrity, CBF, and neuroprotection. Most excitingly, EVs shed by PCs also contain miRNAs that promote angiogenesis, BBB integrity, and neuronal survival. Intravenous or intranasal administration of EVs shed by naive or stimulated PCs could have significant therapeutic effects in conditions such as aging-related disease, AD, PD, peripheral nerve injury, and diabetic retinopathy. Further research on EVs will undoubtedly gain us a better understanding of PC mechanisms and further open new clinical applications.

Although the therapeutic protective effect of PC-EVs derived from different PC sources is reported both in vitro and in vivo, several constraints need to be overcome if progress is to be made toward the therapeutic use of PC-EVs in modulating angiogenesis, BBB integrity, and neuroprotection/neurogenesis. We need standardized isolation and purification methods, protocols for efficient drug loading and safety, and potency and end-product quality control standards. In addition, as the progenitors of PCs are tissue-specific, so is their function [168,169]. In sum, the correct choice of PC-EV source could be challenging. EVs isolated from PCs from different tissue sources may show variable protein, RNA, and lipid content. Thus, the impact of PC-EV heterogeneity on therapeutic efficiency needs to be further revealed. Secondly, PC-EV to recipient cells’ biogenesis, secretion mechanism, and fusion mechanism are still unclear. Understanding such mechanisms would provide a significant opportunity to manipulate their characteristics, composition, and cell interaction to further advance their therapeutic applications. Like other cell-derived EVs, the detrimental effect of the PC-EVs has not been elucidated yet. For instance, EVs have the potential to carry cargoes such as misfolded and mutant proteins related to neurodegenerative disease [170]. Characterizing PC-EVs composition during both physiological and pathological states is beneficial for discerning the progression of neurodegenerative disease. Once these limitations are overcome, PC-EVs could be used as a potential therapeutic agent in vascular and neurological disorders.

## Figures and Tables

**Figure 1 cells-11-03108-f001:**
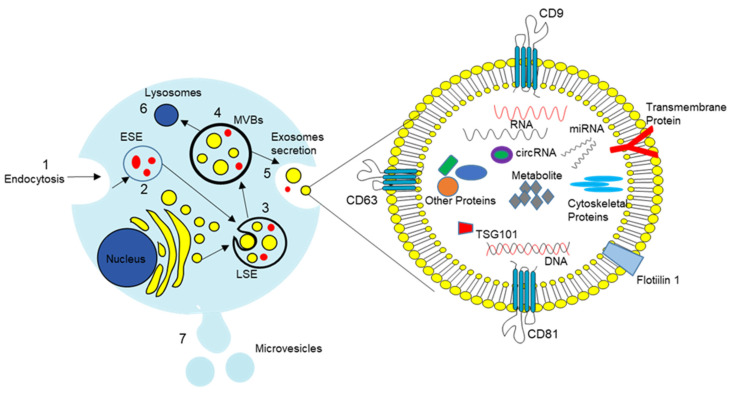
Illustration of the biogenesis of EVs and their cargo. Exosomes, generated by endocytosis (1) of the endosomal membrane, form early (2) and late (3) endosomes and multivesicular bodies (MVBs) (4). After the fusion of the MVBs with the plasma membrane, exosomes are released into the extracellular space (5). MVBs fuse with autophagosomes and are degraded in lysosomes (6). Microvesicles, by contrast, are generated by simple budding of the plasma membrane (7). Exosomes carry proteins, miRNA, mRNA, DNA, and lipids. Tetraspanins (CD9, CD63, and CD81) on the surface serve as markers for EV identification.

**Figure 2 cells-11-03108-f002:**
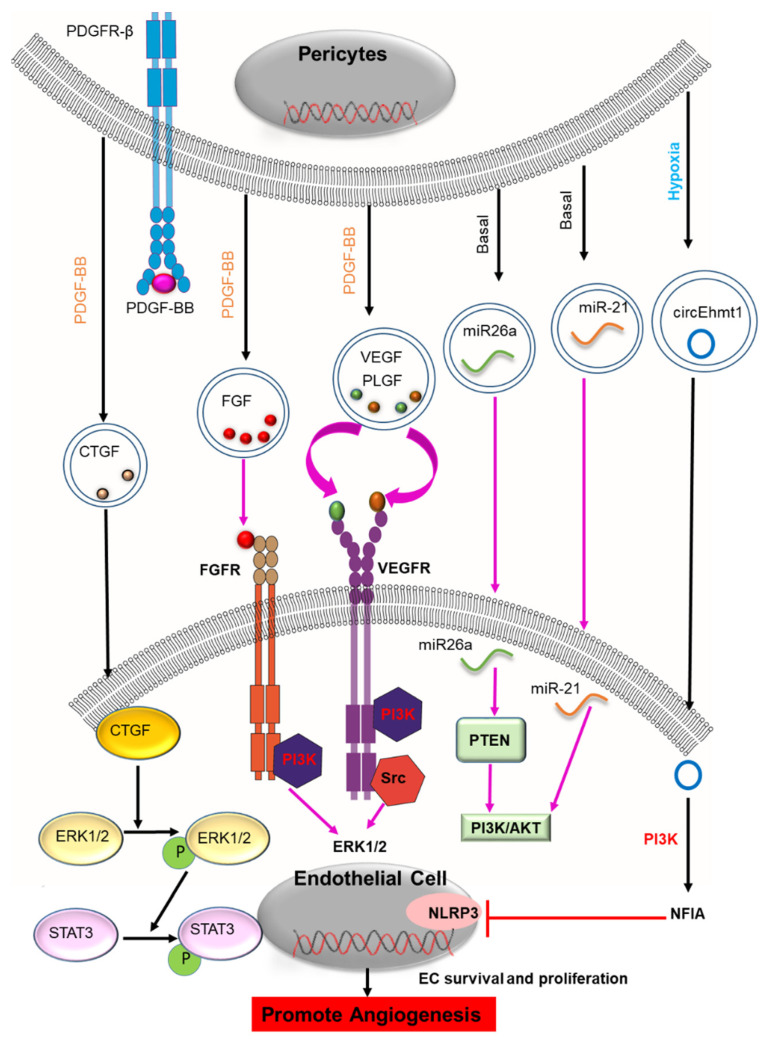
Postulated mechanisms through which PC-EVs are involved in angiogenesis. In response to PDGF-BB, PC-EVs containing several growth factors (CTGF, PLGF, FGF, VEGF) are released. These are transferred to the recipient endothelial cells (ECs) to induce pro-angiogenic signaling in the ECs and promote angiogenesis. CTGF activates the ERK1/2-STAT3 axis in vascular endothelial cells and facilitates the onset of the initial stage of angiogenesis. VEGF and FGF released by platelet-derived EVs act on the VEGF and FGF receptors to exert a pro-angiogenic effect by activating PI3 kinase, src kinase, and ERK1/2. The cargo of EVs also has various miRNAs. Among these, miR26a targets PTEN, activating PI3K/AKT to promote survival and proliferation in the ECs and, thus, angiogenesis. Under hypoxic conditions, PC-released EVs contain circular RNA. Among the circRNAs, circEhmt1 enhances the expression of the transcription factor NFIA while inhibiting the NRLP3 inflammasome, thereby promoting survival and proliferation of the ECs and promoting angiogenesis. The black arrow indicates identified components and mechanism of action of PC-EVs, and the magenta arrow indicates hypothesized Mechanism in PC-EVs based on identified mechanisms from other EVs. (miRNA: an acronym for micro RNA; circRNA: circular RNA; EVs: extracellular vesicles; VEGF: Vascular endothelial growth factor; PLGF: placental growth factor; CTGF: connective tissue growth factor; ERK1/2: extracellular signal-regulated kinase 1 and 2; PI3K: phosphoinositide 3-kinase; PTEN: phosphatase and TENsin homolog deleted on chromosome 10; STAT-3: Signal transducer and activator of transcription).

**Figure 3 cells-11-03108-f003:**
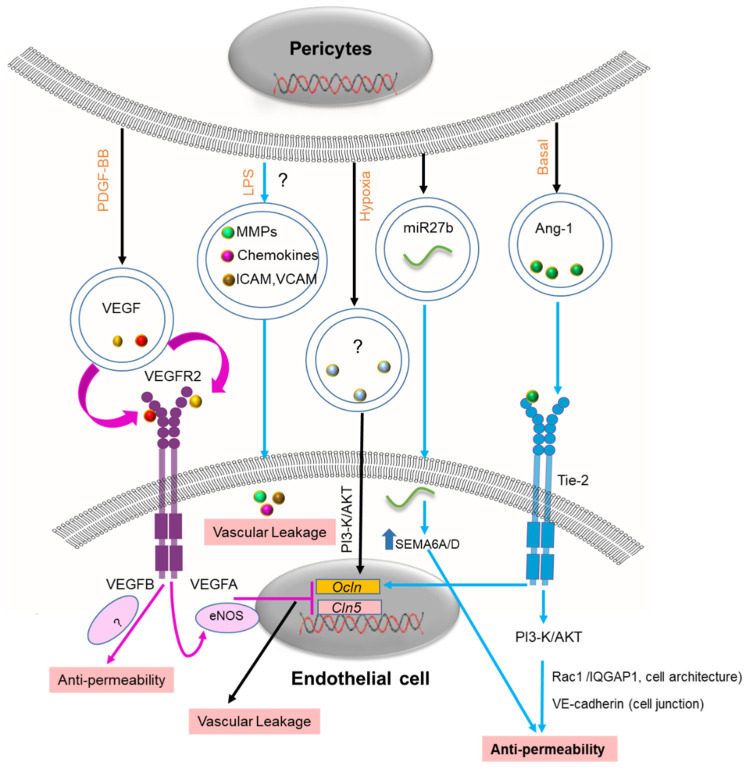
Postulated mechanisms of PC-EVs in the BBB. In response to PDGF-BB, PC-EVs release VEGF to recipient endothelial cells (ECs). VEGF-A inhibits the expression of tight junction proteins such as Ocln and Cln5 through an eNOS signaling pathway, inducing vascular leakage. EVs contain various miRNAs that regulate BBB integrity when transferred to ECs. Among them, miR7b acts by targeting SEMA6A/D, enhancing endothelial barrier function. Hypoxia-induced PC-EV release increases Ocln and Cln5 protein expression by activating the PI3K/Akt pathway. The exact cargo of the PC-EVs has not been reported. Ang-1 released by PC-EVs act on the Tie-2 receptor and promote barrier function by enhancing Ocln gene expression. Ang-1/Tie2 targets the PI3/Akt signaling pathway to promote the expression of cellular junction protein (VE-cadherin), thereby strengthening the cellular architecture. LPS-induced secretion of inflammatory mediators (MMP9, chemokine) and adhesion molecules (ICAM, VCAM) from PCs may promote vascular leakage. PC-EV content released by stimulation with LPS has not yet been reported. The black arrow indicates identified components and mechanism of action of PC-EVs, and the magenta arrow indicates hypothesized Mechanism in PC-EVs based on identified mechanisms from other EVs. The blue arrow indicates the speculative mechanism (miRNA: an acronym for micro RNA; PCs: pericyte; EVs: extracellular vesicles; VEGF: Vascular endothelial growth factor; PI3K: phosphoinositide 3-kinase; Ocln: Occludin; Cln5: Claudin-5; ICAM: intercellular adhesion molecule 1; VCAM: Vascular cell adhesion 1; MMPs: matrix metalloproteinases; Ang-1: angiopoietin-1; BBB: Blood–brain barrier). “?” indicates the content of the PC-EVs is not identified.

**Figure 4 cells-11-03108-f004:**
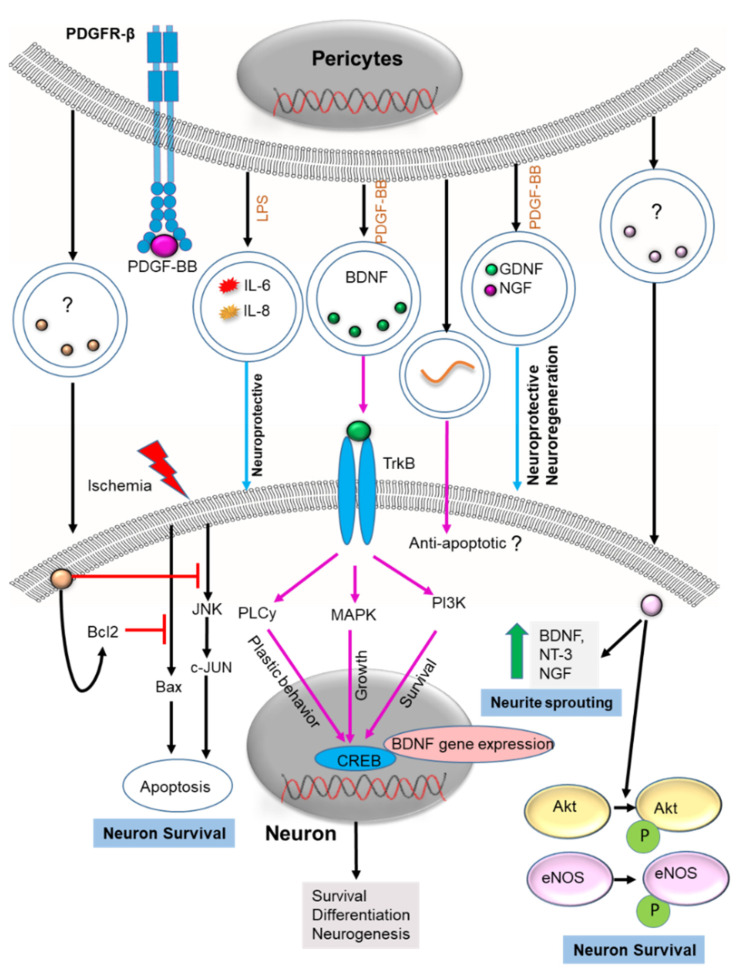
Postulated mechanisms of PC-EVs in neuroprotection and neurogenesis. In response to PDGF-BB, PC-EVs release BDNF to recipient neurons. BDNF targets TrkB receptors and activates PLCγ, MAPK, and PI3K pathways to promote plasticity, growth, and survival. These pathways target CREB transcription factors, enhancing BDNF gene expression and promoting survival, differentiation, and neurogenesis. PC-EV release of growth factors such as GDNF, PTN, and NGF also protects against neurodegenerative pathology. PC-EVs enhance the expression of BcL2 for survival and inhibit both Bax apoptosis and the JNK/jun-c cell death pathway, thus promoting neuronal survival. PC-EVs released from pericytes also increase the expression of growth factors, including BDNF, NT-3, and NGF, which encourage neurogenesis. PC-EV phosphorylated Akt and eNOS promote neuronal survival. “?” indicates the content of the PC-EVs is not identified. Various miRNAs transferred to neurons may also promote neuronal survival and neurogenesis. “?” means the pathway is unknown. The black arrow indicates identified mechanism, the magenta arrow indicates the hypothesized Mechanism in PC-EVs based on the mechanism identified in other EVs, and the blue arrow indicates the speculative mechanism (miRNA: an acronym for micro RNA; PCs: pericyte; EVs: extracellular vesicles; BDNF: Brain-derived neurotrophic factor; NGF: nerve growth factor; GDNF: Glial cell-derived neurotrophic factor; NT-3: Neurotrophin-3; JNK: c-Jun N-terminal kinase; MAPK: Mitogen-activated protein kinase; PI3K: Phosphoinositide 3-kinase; eNOS: endothelial nitric oxide synthase; TrkB: Tropomyosin receptor kinase B; PDGF: Platelet-derived growth factor).

**Figure 5 cells-11-03108-f005:**
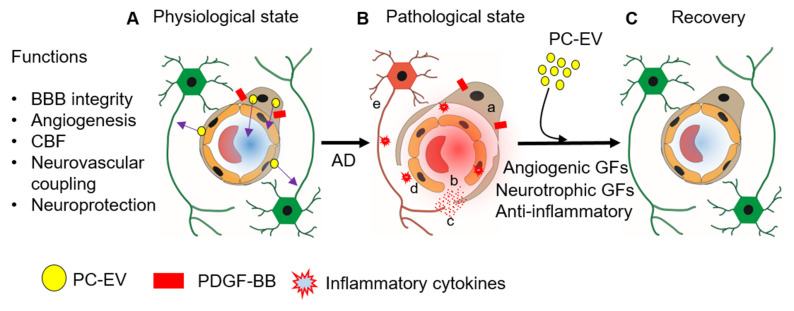
PC-EVs are essential for vascular health, blood–brain barrier integrity, and neuronal health. (**A**) PDGF-BB stimulated the release of PC-EVs to exert effects on neighboring ECs and neurons or through the bloodstream on distant cells. The transport of EVs is key for maintaining vascular and neuronal health. (**B**) Vascular disease associated with neurodegenerative disease, e.g., AD, is caused by, a; pericyte detachment and loss, b; EC death, c; BBB disruption, d; release of inflammatory cytokines, e; neuroinflammation and neuron death. (**C**) Delivery PC-EVs in certain disease conditions could also restore vascular-neural damage and repair the pathology associated with the vascular damage. PCs stimulated with PDGF-BB, for example, release EVs that prevent vascular damage associated with neurodegenerative disease.

**Table 1 cells-11-03108-t001:** PCs-EV cargo under normal and stimulated conditions.

Source of PC-EVs	Culture Condition	PC-EV Cargo	Beneficial or Adverse Effect	Reference
PC-EVs from retinal PCs	Normal	CTGF, CD44	Protected lung tissue and improved pulmonary function EVs protected VEC function	[47]
	Hypoxia	circEhmt1	Protected endothelial cells from HG-induced injury	[48]
	Diabetic and non-diabetic mouse	cPWWP2A	Controlled diabetes mellitus-induced microvascular dysfunction	[49]
PC-EVs in brain	Stimulated with PDGF-BB	BDNF, bFGF, BNGF, VEGF, PLGF	EV release of BDNF, bFGF, bNGF, VEGF, PLGF Neuroprotection, angiogenesis	[41,46]
	Stimulated with LPS	IL-6, IL-8, IL-10, MCP-1,	Vascular and neuronal inflammation	[41]
PC-EVs in brain	Normal	miR-26a, miR-143, miR 122-5p, miR-6240, miR-122-5p, miR-181a, miR-21, miR-29a, let-7c-5p, let-7a-5p miR-1285	Biomarkers and treatment of hypertension	[43]
PC-EVs in muscle	Stimulation with H_2_O_2_	antioxidants and anti-inflammatory compounds	Anti-oxidative Anti-inflammatory Extracellular matrix remodeling	[50]

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
