# Peer review of "The Emerging Role of Pericyte-Derived Extracellular Vesicles in Vascular and Neurological Health"

_cells, 2022, doi:10.3390/cells11193108_

Round 1

Reviewer 1 Report

This is a marvelous review. However, the role of PC-exosomes in diseases should be explained in greater detail

Author Response

Reviewer 1

This is a marvelous review. However, the role of PC exosomes in diseases should be explained in greater detail.

Response: Reviewer #1’s comments are greatly appreciated. Yes, we agree with the reviewer that more detailed information on PC-EVs is better. Unfortunately, studies on PC-EVs are very recent, and their report is still quite limited. We are much in the initial discovery phase. We have revised the text to emphasize the limitations on page 17.

Reviewer 2 Report

The field of biology of extracellular vesicles is of high importance and rapidly developing, with more than 1000 papers being published per year. The novel aspect of the review paper by Kushal Sharma and colleagues is that their attention was focused on pericyte-derived EVs. In general, the review is well written, examines recent literature on structure of EV, functions in vascular and nervous system pathophysiology. I would suggest that the review would benefit if authors addressed few questions that are not widely discussed in other reviews:

1. What is the fate of EVs in the extracellular environment? How they are recognized by target cells? Are all EVs taken up into target cells by macroendocytosis or other mechanisms?

2. It is described that stimulated pericytes release various growth factors which are supposed to act via receptors on the outer side of cellular membranes. But growth factors are enclosed in phospholipid membranes in EV which would prevent interaction with receptors. Thus, how interaction of EV growth factors with receptors occurs? Can EVs release their content outside of cells and by which mechanism?

3. As shown in Fig. 1, EVs contain variety of molecules including proteins, DNA, RNR, etc. If EV can release their content into extracellular environment, wouldn‘t it be dangerous to have these various molecules released?

Author Response

Reviewer 2

The field of biology of extracellular vesicles is of high importance and rapidly developing, with more than 1000 papers being published per year. The novel aspect of the review paper by Kushal Sharma and colleagues is that their attention was focused on pericyte-derived EVs. In general, the review is well written, examines recent literature on structure of EV, functions in vascular and nervous system pathophysiology. I would suggest that the review would benefit if authors addressed few questions that are not widely discussed in other reviews:

  1. What is the fate of EVs in the extracellular environment? How are they recognized by target cells? Are all EVs taken up into target cells by macroendocytosis or other mechanisms?

Response: Great questions!! We thank Reviewer #2 for the time and effort put into reviewing our manuscript.  The questions raised by reviewer #2 are now addressed on page 4, Section 1.2, Line 135: 162. We restate below:

EVs are distributed in different tissues, organs, and biological fluids both locally or systemically [1]. Systemic exosomes are rapidly cleared from the blood circulation by macrophages, neutrophils, and endothelial cells and are transported to the liver, spleen, lungs, and gastrointestinal tract [2, 3]. EV access to tissue involves multiple cellular uptake and release cycles [4]. However, different factors such as cellular origin, membrane composition, EV size, and pathological conditions in the host may affect how EVs are transported [5].

 Although all cell types share in non-specific uptake of exosomes [6], specific targeting of recipient cells is required to deliver specific cargo and exert specific function [7]. The surface composition of the exosome mediates this delivery specificity. EV-target cell interactions involve tetraspanins, integrins, ECM proteins, immunoglobulin superfamily members, proteoglycans, heat shock proteins, and lectins [8]. For example, heat shock protein 70 predominantly clusters around the exosomal membrane [9]. There is evidence that it interacts with Toll-like receptor 4 to initiate a critical signaling cascade necessary for neuron survival [10]. Umbilical cord blood-derived exosomes express tumor antigens such as MHC-I, MHC-II, and tetraspanins (CD34, CD80) and stimulate T cell proliferation to produce antitumor activity [11].

EVs then interface with neighboring cells through different modes, including receptor-binding without internalization, phagocytosis, macropinocytosis, internalization by clathrin-caveolae- and lipid raft-mediated endocytosis, filopodia-based endocytosis, and bursting in the acidic environment [12]. If internalized, the EVs release their cargo in the cytoplasm, and the internalized cargo may regulate the cell at the transcription or translation level [13, 14]. EVs carry a range of cargo for deposition into recipient cells, from genetic materials (RNA, DNA, and miRNA), to proteins, lipids, and transcription factor receptors. They can effectively alter biological response. The exosome-mediated response could be either therapeutic or disease-promoting [15]. While it is clear exosomes are a promising therapeutic tool for treatment of different pathologies, it is clear they are also associated with development of diseases such as diabetes, Alzhemier’s, progression of cancer, and inflammatory disease [16].

  1. It is described that stimulated pericytes release various growth factors which are supposed to act via receptors on the outer side of cellular membranes. But growth factors are enclosed in phospholipid membranes in EV which would prevent interaction with receptors. Thus, how interaction of EV growth factors with receptors occurs? Can EVs release their content outside of cells and by which mechanism?

Response: We thank the reviewer for this insightful comment. We have addressed this issue on page 8, Line 285-293, restated below:

Recent studies of VEGFs indicate that VEGF isoforms are bound to the surface of small EV (sEVs) through heparin surface binding or bound to microvesicles through heat shock protein 90 (HSP90) [17, 18]. Microvesicle-associated VEGF-HSP90 has been shown to stimulate VEGFR2 phosphorylation in endothelial cells and tube formation [19]. EV-associated VEGF189 has also been shown to stimulate cellular domain of VEGFR2 and induce phosphorylation. The VEGFR2 signaling elicited in endothelial cells then increases microvessel density in vivo. This finding supports the idea that VEGF in PC-EVs may interact with the extraceullar domain of VEGFR2 and the VEGFR signaling induces angiogenesis.

  1. As shown in Fig. 1, EVs contain variety of molecules including proteins, DNA, RNR, etc. If EV can release their content into extracellular environment, wouldn‘t it be dangerous to have these various molecules released?

Response: Yes, EVs can release their cargo into the extracellular environment and the result could be either therapeutic or disease promoting. We have briefly mentioned this issue in page 4, Line 156- 162, restated below:

EVs carry a range of cargo for deposition in recipient cells, from genetic materials (RNA, DNA, and miRNA), proteins, lipids, and transcription factor receptors. They can effectively alter biological response. The exosome-mediated response could be either therapeutic or disease-promoting [15]. While it is clear exosomes are a promising therapeutic tool for treatment of different pathologies, it is clear they are also associated with development of diseases such as diabetes, Alzhemier’s, progression of cancer, and inflammatory disease [16].

References:

  1. Gurung, S., et al., The exosome journey: from biogenesis to uptake and intracellular signalling. Cell Communication and Signaling, 2021. 19(1): p. 47.
  2. Kim, S.H., et al., Effective treatment of inflammatory disease models with exosomes derived from dendritic cells genetically modified to express IL-4. J Immunol, 2007. 179(4): p. 2242-9.
  3. Charoenviriyakul, C., et al., Cell type-specific and common characteristics of exosomes derived from mouse cell lines: Yield, physicochemical properties, and pharmacokinetics. European Journal of Pharmaceutical Sciences, 2017. 96: p. 316-322.
  4. Lakhal, S. and M.J. Wood, Exosome nanotechnology: an emerging paradigm shift in drug delivery: exploitation of exosome nanovesicles for systemic in vivo delivery of RNAi heralds new horizons for drug delivery across biological barriers. Bioessays, 2011. 33(10): p. 737-41.
  5. Kooijmans, S.A.A., O.G. de Jong, and R.M. Schiffelers, Exploring interactions between extracellular vesicles and cells for innovative drug delivery system design. Advanced Drug Delivery Reviews, 2021. 173: p. 252-278.
  6. Zech, D., et al., Tumor-exosomes and leukocyte activation: an ambivalent crosstalk. Cell Communication and Signaling, 2012. 10(1): p. 1-17.
  7. Horibe, S., et al., Mechanism of recipient cell-dependent differences in exosome uptake. BMC Cancer, 2018. 18(1): p. 47.
  8. Buzás, E.I., et al., Molecular interactions at the surface of extracellular vesicles. Semin Immunopathol, 2018. 40(5): p. 453-464.
  9. Chanteloup, G., et al., Membrane-bound exosomal HSP70 as a biomarker for detection and monitoring of malignant solid tumours: a pilot study. Pilot and Feasibility Studies, 2020. 6(1): p. 35.
  10. Müller, U., Exosome-mediated protection of auditory hair cells from ototoxic insults. J Clin Invest, 2020. 130(5): p. 2206-2208.
  11. Guan, S., et al., Umbilical cord blood-derived dendritic cells loaded with BGC823 tumor antigens and DC-derived exosomes stimulate efficient cytotoxic T-lymphocyte responses and antitumor immunity in vitro and in vivo. Cent Eur J Immunol, 2014. 39(2): p. 142-51.
  12. Mulcahy, L.A., R.C. Pink, and D.R.F. Carter, Routes and mechanisms of extracellular vesicle uptake. Journal of extracellular vesicles, 2014. 3: p. 10.3402/jev.v3.24641.
  13. Abels, E.R. and X.O. Breakefield, Introduction to Extracellular Vesicles: Biogenesis, RNA Cargo Selection, Content, Release, and Uptake. Cellular and molecular neurobiology, 2016. 36(3): p. 301-312.
  14. Berumen Sánchez, G., et al., Extracellular vesicles: mediators of intercellular communication in tissue injury and disease. Cell Communication and Signaling, 2021. 19(1): p. 104.
  15. Kalluri, R. and V.S. LeBleu, The biology, function, and biomedical applications of exosomes. Science, 2020. 367(6478).
  16. Engin, A., Dark-Side of Exosomes. Adv Exp Med Biol, 2021. 1275: p. 101-131.

Reviewer 3 Report

The topic of the Review by Sharma and co-authors is novel and significant. The review is comprehensive and discusses recent literature. However, I have a few concerns and suggestions for the authors. 

1 – Section 1.1. The authors should better discuss the limitations of pericyte identification and targeting (using Flox-Cre approaches) compared to smooth muscle cells (SMC) and mesenchymal stem cells (MSC).

2 – The authors described different types of extracellular vesicles in Section 1.2. However, the authors use the term “EV” to refer to all types of micro-vesicles but mainly talking about exosomes in their review. I feel that the authors can be more specific when they speak about particles of a specific size.

3 – Figures 2, 3, and 4. Some parts of these schematics are seen as too speculative. For example, experimental data does not support the release of growth factors, such as VEGF, PLGF, CTGF, Ang1, BDNF, from perycite-derived EVs. I recommend being clear on the schematics (e.g., highlight with different color and have an annotation in the legend) of which mechanisms are supported by experiments versus hypotheses and speculations.   

4 – About half of the recent literature (for the last five years) cited in the review represents reviews from other authors and not the original research articles.

5 – Para 87 has a typo in word exomeres.

Author Response

Reviewer 3

The topic of the Review by Sharma and co-authors is novel and significant. The review is comprehensive and discusses recent literature. However, I have a few concerns and suggestions for the authors. 

  1. Section 1.1. The authors should better discuss the limitations of pericyte identification and targeting (using Flox-Cre approaches) compared to smooth muscle cells (SMC) and mesenchymal stem cells (MSC).

Response: We thank Reviewer #3 for insightful comments. The issue raised by reviewer #3 is now addressed in the revised version. Location: page 2, Section 1.1; PCs Line: 65-80, as restated below:

Taking advantage of the Cre-Lox systems, pericyte function can be studied by knocking in or out the gene within the pericyte [1]. In addition, the combined technique of optogenetic with 2-photon imaging has been shown to stimulate pericytes to generate a state of vessel constriction, eliciting pericyte-driven capillary diameter changes. However, this has been inconsistent in all studies [2], possibly due to the non-specificity of the Cre-reporter line or model/and or light source used. For instance, genetic mouse model such as NG2- and PDGFR-β reporter mice not only target the pericytes but also VSMC [3] or fibroblast [4]. A better genetic mouse model must be design to target and validate the pericyte function

  1. The authors described different types of extracellular vesicles in Section 1.2. However, the authors use the term “EV” to refer to all kinds of micro-vesicles but mainly talk about exosomes in their review. I feel that the authors can be more specific when they speak about particles of a specific size.

Response: We apologize for the inconsistency. We have now changed the word “exosomes” into “EV” wherever it is necessary.

  1. Figures 2, 3, and 4. Some parts of these schematics are seen as too speculative. For example, experimental data does not support the release of growth factors, such as VEGF, PLGF, CTGF, Ang1, and BDNF, from perycite-derived EVs. I recommend being clear on the schematics (e.g., highlight with different colors and have an annotation in the legend) of which mechanisms are supported by experiments versus hypotheses and speculations.  

Response: Thanks to the reviewer for pointing out this. We have now used a different color to describe actual, speculative, and hypothetical mechanisms by adding an annotation in the legend to clarify this.

  1. About half of the recent literature (for the last five years) cited in the review represents reviews from other authors and not the original research articles.

Response: Reference from the original research article has been added.

  1. Para 87 has a typo in word exomere

Response: The typo in line 87 is now corrected in Line 107 of the revised manuscript.

References

  1. Ma, Q., et al., Blood-brain barrier-associated pericytes internalize and clear aggregated amyloid-β42 by LRP1-dependent apolipoprotein E isoform-specific mechanism. Molecular neurodegeneration 2018, 13, p. 1-13.
  2. Hill, R.A., et al., Regional blood flow in the normal and ischemic brain is controlled by arteriolar smooth muscle cell contractility and not by capillary pericytes. Neuron 2015, 87, p. 95-110.
  3. Jung, B., et al., Visualization of vascular mural cells in developing brain using genetically labeled transgenic reporter mice. Journal of Cerebral Blood Flow & Metabolism 2018, 38, p. 456-468.
  4. Alex, L., et al., Validation of Specific and Reliable Genetic Tools to Identify, Label, and Target Cardiac Pericytes in Mice. Journal of the American Heart Association 2022, 11, p. e023171.

Reviewer 4 Report

In the manuscript (Journal Cells , Manuscript ID cells-1909420, Type Review. Title The emerging role of pericyte-derived extracellular vesicles in vascular and neurological health Authors Kushal Sharma, Yunpei Zhang, Keshav Raj Paudel, Allan Kachelmeier, Phillip M. Hansbro, Xiaorui Shi, Section Organelle Function Special Issue Extracellular Vesicles: Potential Roles in Regenerative Medicine, the authors review the pericyte-derived extracellular vesicles including their formation, identification, cargo (e.g. growth factors, inflammatory cytokines, ...), physiology, as well as, their therapeutic potential in some diseases such as Parkinson;s, and spinal cord injuries, erectile

dysfunctions, diabetic retinopathies and muscle atrophy.

The work is well systematized and overall interesting.

I have no major considerations. The authors should highlight the limitations and, if possible, could enrich the schemes with some images of pericytes and their vesicles.

Author Response

Reviewer 4

In the manuscript (Journal Cells , Manuscript ID cells-1909420, Type Review. Title The emerging role of pericyte-derived extracellular vesicles in vascular and neurological health Authors Kushal Sharma, Yunpei Zhang, Keshav Raj Paudel, Allan Kachelmeier, Phillip M. Hansbro, Xiaorui Shi, Section Organelle Function Special Issue Extracellular Vesicles: Potential Roles in Regenerative Medicine, the authors review the pericyte-derived extracellular vesicles including their formation, identification, cargo (e.g. growth factors, inflammatory cytokines, ...), physiology, as well as, their therapeutic potential in some diseases such as Parkinson;s, and spinal cord injuries, erectile

dysfunctions, diabetic retinopathies and muscle atrophy.

The work is well systematized and overall interesting.

I have no major considerations. The authors should highlight the limitations and, if possible, could enrich the schemes with some images of pericytes and their vesicles.

Response: We thank Reviewer for suggestions. The limitation has now been highlighted under Section 5, Conclusion and future perspective ; Line 631-649. Restated below:

“Although the therapeutic protective effect of PC-EVs derived from different PC sources is reported both in-vitro and in-vivo, several constraints need to be overcome if progress is to be made toward the therapeutic use of PC-EVs in modulating angiogenesis, BBB integrity, and neuroprotection/neurogenesis. We need standardized isolation and purification methods, protocols for efficient drug loading and safety, and potency and end-product quality control standards. In addition, as the progenitors of PCs are tissue-specific, so is their function [1, 2]. In sum, the correct choice of PC-EV source could be challenging. EVs isolated from PCs from different tissue sources may show variable protein, RNA, and lipid content. Thus the impact of PC-EV heterogeneity on therapeutic efficiency needs to be further revealed. Secondly, PC-EV to recipient cells' biogenesis, secretion mechanism, and fusion mechanism are still unclear. Understanding such mechanisms would provide a significant opportunity to manipulate their characteristics, composition, and cell interaction to further advance their therapeutic applications. Like other cell-derived EVs, the detrimental effect of the PC-EVs has not been elucidated yet. For instance, EVs have the potential to carry cargoes such as misfolded and mutant proteins related to neurodegenerative disease [3]. Characterizing PC-EV composition during both physiological and pathological state is beneficial for discerning the progression of neurodegenerative disease. Once these limitations are overcome, PC-EVs could be used as a potential therapeutic agent in vascular and neurological disorders”.

Reference:

  1. Chen, W.C., et al., Human myocardial pericytes: multipotent mesodermal precursors exhibiting cardiac specificity. Stem cells 2015, 33, p. 557-573.
  2. Zhu, S., et al., Versatile subtypes of pericytes and their roles in spinal cord injury repair, bone development and repair. Bone Research 2022, 10, p. 30.
  3. Lee, J.Y. and H.S. Kim, Extracellular Vesicles in Neurodegenerative Diseases: A Double-Edged Sword. Tissue Eng Regen Med 2017, 14, p. 667-678.

Reviewer 5 Report

At a moment, pericytes (PC) are attracted a considerable attention in the field of regenerative medicine.  As a low committed progenitors of mesenchymal lineage, these cells manifest high secretory activity. The present MS is devoted to the presentation of current state-of-art in C extra-vesicles (EV)s’ features and  the contribution of EV path in realization of  PC activity.

In general, the topic of the Review by Sharma and co-authors is highly significant. The review is nicely written and discusses recent literature. However, I have a few questions for the authors.  

 Line 23. The authors have sited [23] indicating that PC   are pluripotent stem cell-like cells.   This is a controversial statement.  The trans-differentiation in other than mesenchymal lineages have been demonstrated in vitro only. Please specify.

1.3.2. PC-EV cargo. Describing the PC-EV features, did the authors cite only in vitro observations of the data from in vivo studies are also available?

As authors indicate, EVs are secreted by different cell types including PC cognate MSCs. Are there studies of comparative effectiveness of EVs from different mesenchymal stromal cells as MSCs, fibroblasts, more committed progeny?  

Some minor spell-check and grammar-check are recommended.

Author Response

Reviewer 5

At a moment, pericytes (PC) are attracted a considerable attention in the field of regenerative medicine.  As a low committed progenitors of mesenchymal lineage, these cells manifest high secretory activity. The present MS is devoted to the presentation of current state-of-art in C extra-vesicles (EV)s’ features and  the contribution of EV path in realization of  PC activity.

In general, the topic of the Review by Sharma and co-authors is highly significant. The review is nicely written and discusses recent literature. However, I have a few questions for the authors. 

  1. Line 23. The authors have sited [23] indicating that PC are pluripotent stem cell-like cells.  This is a controversial statement. The trans-differentiation in other than mesenchymal lineages have been demonstrated in vitro only. Please specify.

Response: Thank you for your advisive comment and suggestion. We have now corrected the text to be more accurate. Please see page 2, Line 86-87:  statement below:

More recently, pericyte-like cells could be derived from human pluripotent stem cells.”

 Yes the transdifferentiatiion has been observed in both in-vivo and in-vitro. We have added references to support this fact in Line 88:

  1. 3.2. PC-EV cargo. Describing the PC-EV features, did the authors cite only in vitro observations of the data from in vivo studies are also available?

Response: The reviewer is correct. This manuscript mentioned observation from in-vitro characterization of the PC-EV cargo. Study on PC-EV is recent and only the in-vitro characterization of the PC-EV has been done so far.  We have addressed this as our limitation in our study in section 5; Conclusion and Future Perspective; Line 631-649

  1. As authors indicate, EVs are secreted by different cell types including PC cognate MSCs. Are there studies of comparative effectiveness of EVs from different mesenchymal stromal cells as MSCs, fibroblasts, more committed progeny?  

Response: Yes, there is a study  doing comparative effectiveness of EVs where induced pluripotent stem cells showed better therapeutic effect on corneal epithelial defect than mesenchymal stem cells-derived exosomes  [1]. However,  the comparative study with PC-EV has not been elucidated yet and hopefully further investigation is necessary to measure the effectiveness of PC-EV. 

  1. Some minor spell-check and grammar-check are recommended.

Response: We have now corrected all grammatical errors.

References:

  1. Wang, S., et al., Comparison of exosomes derived from induced pluripotent stem cells and mesenchymal stem cells as therapeutic nanoparticles for treatment of corneal epithelial defects. Aging (Albany NY), 2020. 12(19): p. 19546-19562.

Round 2

Reviewer 2 Report

All questions raised were answered, additional discussion included. I have no more comments.